# Bird Health, Housing and Management Routines on Swedish Organic Broiler Chicken Farms

**DOI:** 10.3390/ani10112098

**Published:** 2020-11-11

**Authors:** Lina Göransson, Jenny Yngvesson, Stefan Gunnarsson

**Affiliations:** Department of Animal Environment and Health, Swedish University of Agricultural Sciences (SLU), P.O.B. 234, S-53223 Skara, Sweden; jenny.yngvesson@slu.se (J.Y.); stefan.gunnarsson@slu.se (S.G.)

**Keywords:** slower-growing, welfare, foot pad dermatitis, hock burns, plumage, gait, body weight uniformity, mortality

## Abstract

**Simple Summary:**

Knowledge of bird welfare and management on Swedish organic broiler farms is limited, since the number of farms began to increase only recently when two slower-growing hybrids became commercially available in Sweden. The aim of the study was to obtain information about chicken health and other welfare aspects, along with details of housing and management routines, in order to increase the knowledge and describe the current situation of these farms. Clinical examinations revealed no severe remarks, however minor to moderate plumage dirtiness, food pad dermatitis and hock burns were found in 47%, 21% and 13% of the birds, respectively. Higher body weights were significantly correlated to an increased prevalence of hock burns and dirty plumages. Although no severe walking impairments were observed, minor to moderate gait abnormalities were seen in almost two-thirds of all birds assessed. Gait in chickens assessed outdoors was significantly better than in those observed indoors. Flock body weight uniformity was low in all flocks. The study provides new knowledge of two slower-growing hybrids on Swedish organic farms. Further research should be focused on investigating other important aspects related to bird welfare, such as the low flock body weight uniformity and the high mortality rates observed.

**Abstract:**

Slower-growing broilers on organic farms have replaced fast-growing hybrids to increase bird welfare. Due to limited knowledge of broiler welfare and management on organic farms in Sweden, the study aim was to gather information regarding health, housing and management routines, in order to describe the current situation on these. Farm visits performed in 2018 included 8 out of 12 established organic farms, on which either Rowan Ranger or HubbardJA57/HubbardJA87 were reared. Chickens in the observed flocks were 55 ± 6 (44–62) days of age. Observations included farmer interviews, indoor environment assessments, clinical examinations and gait scoring. Clinical examinations revealed no severe remarks, however minor to moderate plumage dirtiness, food pad dermatitis and hock burns were found in 47%, 21% and 13% of the birds, respectively. Although no severe walking impairments were observed, minor to moderate gait abnormalities were seen in two-thirds of the birds. Gait in birds assessed outdoors was significantly better than in birds observed indoors. Body weight uniformity was low in all flocks. This study provides increased knowledge of certain chicken health and welfare aspects, housing and management on Swedish organic farms. Future research should further investigate important aspects related to bird welfare, such as the high mortality rates observed.

## 1. Introduction

The concept of animal welfare encompasses the biological functioning and health of the animal, natural living and the possibility to express natural behaviour, as well as the subjective experience of the individual [1]. It has been proposed as a fourth dimension of sustainable animal farming [2], alongside the environmental, economic and social dimensions, and is one of the fundamental principles in organic agriculture [3]. Organic animal farming should thus comply with high animal welfare standards and promote animal health and well-being, with emphasis on species-specific behavioural needs [4]. EU regulations on organic broiler production require lower stocking densities and outdoor access for the birds [4], so as to better provide them with the opportunity to perform species-specific natural behaviours such as foraging and dust bathing. Chickens must also be provided with roughage during periods without outdoor access [5]. Natural light is obligatory and, when complemented by artificial light, a minimum period of eight consecutive hours without the latter is required for a nocturnal rest period [5]. To enable the best animal health possible under the prevailing production conditions, suitable hybrids should be selected for the purpose [4]. Fast-growing broilers have therefore been replaced by slower-growing hybrids [5], as the welfare of the latter and their suitability for organic production appear better [6,7,8].

The slower-growing hybrid Rowan Ranger^®^ [9] became commercially available in Sweden during 2014, followed by Hubbard^®^ [10] strains in 2016. In 2018, only 0.8% of total Swedish broiler production was organic [11]. The genetic variation within hybrids is a current problem on organic farms, making management and production unpredictable, as individuals within flocks tend to differ widely in body weight [12]. Moreover, organic poultry farming may involve animal welfare issues such as predation [13,14]. Numerous studies show that, despite their physical ability, only a small proportion of the flock ranges when given outdoor access and most chickens remain in close proximity to the house [15,16,17,18]. A number of studies have been situated in other European countries [6,17,18,19], but the majority of these seem not to have been performed under commercial settings, or under conditions compatible with those in Northern Europe.

Organic broiler production in Sweden at the time of the study was rather recent, and knowledge of bird welfare, production and management on farms is still limited. The aim of this empirical study was thus to collect information and accumulate knowledge of broiler health and other welfare aspects, along with details of housing and management routines, in order to describe the present situation on these farms. The aim further included the identification of areas of relevance for future research on organic broiler farms in Sweden.

## 2. Materials and Methods

This study comprised behavioural observations and clinical scoring of commercial broiler chickens without any invasive treatment, and thus ethical approval by an ethics committee for animal experiments was not required according to Swedish legislation [20].

### 2.1. Farm Visits

Based on an inventory of the current organic broiler farms in Sweden in 2018, the owners of these 12 farms were contacted by telephone. All eight farmers willing to participate in the study did so, with no particular selection of farms to be included in the study. These eight commercial organic broiler farms, all located in the southern third of Sweden, were visited during October (autumn) 2018.

Chickens in the observed flocks were 55 ± 6 (44–62) days of age at the time of the farm visits, which were performed as late in the production cycle and as close to slaughter as possible. The visits were completed on one day each, between 09.30–10.00 h and 15.00–15.30 h, by the first author and one assisting person. One farm was visited from 08.00 h for logistical reasons. All farmers were interviewed according to a structured protocol (Appendix A) covering management and husbandry routines, housing, bird health and behaviour, productivity, and free-range characteristics and utilisation. The farmers also received an e-mail questionnaire with open questions regarding their perceptions and opinions of organic broiler production. The farmers were contacted once more after the visits, to obtain abattoir records on the flocks observed during farm visits.

#### 2.1.1. Housing and Indoor Environment

On each farm, indoor observations were made in one flock of broilers, in one rearing compartment, immediately after the interview with the farmer. If there was more than one flock of suitable age, the farmer selected the flock to be observed. On entering the rearing compartment, a piece of black paper was placed in an elevated position to assess the dust level, on a scale from a (none) to e (paper colour not visible), according to the Welfare Quality^®^ assessment protocol for poultry [21]. Litter quality (Welfare Quality^®^) was then assessed on a scale from 0 (completely dry and flaky) to 4 (sticks to boots once compacted crust is broken) at five standardised locations in the rearing compartment, and the proportion of birds panting or huddling (Welfare Quality^®^) was recorded. These five locations were: “at the entry door”, “below drinker (centre of rearing compartment)”, “at pop-hole (centremost)”, “halfway along outer short side” and “halfway along inner long side”. Observations were always made in this order when walking through the rearing compartment. Dimensions of house and veranda (a roofed platform at ground level with three walls and one curtain, littered floor and natural ventilation, adjacent to the rearing compartment and connecting this to the free-range area) and number and location of windows, pop-holes, drinkers, feeders and indoor environmental enrichment were also recorded.

#### 2.1.2. Gait Scoring and Clinical Examination

Five birds were gait-scored at each of the five locations (*n* = 25), using a six-point scale (Table 1). Completing a full rotation, the observer made five slow turns at each location. Following each turn, starting with the bird closest to the observer and counting to seven, one bird was randomly chosen and then gait-scored as the flock was gently encouraged to move. Gait scoring of an additional 25 individuals in the free-range area was then performed from the veranda, since walking in the free-range visible to the birds tended to scare the animals, hampering accurate observations. Birds were randomly chosen by counting to seven repeatedly, as during indoor gait scoring, but scanning from the left to the right side of the free-range area. If no birds were observed outdoors, only 25 individuals indoors were assessed. Birds on the veranda were not gait-scored.

In each flock, 50 birds were clinically examined. Individual handling of the chickens was done last during indoor observations. While walking slowly around the entire rearing compartment, making a complete circuit, groups of birds were confined until 50 animals had been examined. Approximately 3–15 birds at a time were confined against the wall using three connected compost grids. The birds were gently picked up one at a time, placed in a bucket with a lid and weighed, and thereafter clinically examined (Table 1).

The birds were marked with a black marker pen on one leg before release, to avoid examining the same bird twice. The assessor was trained in gait scoring and clinical assessment of chickens during a farm visit prior to data collection, which was further complemented by video materials for the purpose. The same person performed all clinical examinations and gait scoring of birds throughout the study.

### 2.2. Statistical Analyses

The Welfare Quality^®^ assessment protocol was used for scoring certain welfare parameters in the study, however no indices were calculated according to the protocol. Data compilation and diagram creation were performed in Microsoft Excel (2016). All statistical analyses were performed in R [23]. Results are presented as mean value with standard deviation (min-max), unless otherwise stated. Results were considered significant when *p* < 0.05.

Fisher’s exact test was used to analyse any dependence between the different locations in the rearing compartments and litter quality scores, as well as between floor heating and litter quality scores. The effect of litter quality on foot pad dermatitis (FPD) and hock burns (HB) was analysed through a logistic regression model. FPD scores were transformed into a binary variable (i.e., scores ≥ 1 were pooled), since scores ≥ 2 were few, before the analysis. Litter quality and body weight were included in the model. There was no interaction between these variables, hence interaction was not included in the model. For this analysis, the median of the five separate litter quality scores on each farm was used. Because litter quality scores were directly linked to the individual farms, farm was not included in the model.

The dependence between FPD and HB, between FPD and plumage cleanliness, and between HB and plumage cleanliness was analysed using Fisher’s exact test. The association between body weight (BW) and age, respectively, and HB, FPD, and plumage cleanliness, respectively, was analysed using a logistic regression model. FPD and plumage cleanliness scores were transformed into binary variables (i.e., scores ≥ 1 were pooled), since scores ≥ 2 were few, before the analysis. Hybrid was included as a fixed factor, including the interaction between body weight and age, respectively, and hybrid. Since the participating farms comprised the majority of farms in the country and no randomised selection had been made, farm was also included as a fixed factor, as opposed to as a random factor.

Gait scores (GS) of birds observed indoors and outdoors were compared using Fisher’s exact test. Due to the variations in age and average flock BW, which could largely impact GS, the two farms on which no birds were assessed outdoors were excluded from the analysis comparing indoor and outdoor gait scores. Fisher’s exact test was also used to analyse the effect of BW on indoor GS (indoor scores only, since birds were not observed outdoors in all flocks). Due to the lack of data on individual BW, average flock BW was used for the analysis. There were two distinct groups with reference to average flock BW, one with heavier birds and one with lighter, which were used for comparison. The correlation between age and gait scores was analysed using a logistic regression model. Farm and hybrid were included as fixed factors, including the interaction between age and the latter.

Flock body weight uniformity was calculated according to Toudic (2007) [24], and compared between the two hybrids using the Welch two-sample *t*-test.

## 3. Results

All farms in the study were affiliated with the Swedish private organic incorporated association KRAV^®^ [25], and certified according to KRAV standards [26]. On seven out of the eight farms studied, organic chicken production was established in 2012–2016 without prior chicken production on the farm. The remaining farm had previously had conventional broiler production before conversion to organic production in 2015. Only four of the eight farmers responded to the questionnaire regarding their opinions and perceptions of organic broiler production, thus this information was not included in any further analysis.

### 3.1. Animals and Farm Management

Rowan Ranger (RR) or Hubbard JA57/Hubbard JA87 (H), both slower-growing hybrids, were raised in mixed-sex flocks on five and three farms, respectively. The latter received day-old chicks (RR), whereas the former received eggs (H) for on-farm hatching. All farms had specific arrival compartments in which chicks were kept until around three weeks of age. Eggs were delivered to farms on Friday afternoons, following 18 days of incubation at a commercial hatchery. The farmers reported that hatching most commonly commenced during the following Saturday, or at most within five days after arrival of the eggs.

Artificial lighting was supplied continuously throughout the hatching process, as was it at arrival of the day-old chicks. A nocturnal period was thereafter induced within 3 ± 3 (1–7) days, through either an immediate (two farms) or gradual (six farms) decrease in artificial lighting. In the latter (specific information available from three farms only), this gradual decrease was by one or two hours per day. Initial temperature in the arrival compartments was 34.4 ± 1.5 (32–36) °C, which was gradually decreased to 22.9 ± 2.4 (20–26) °C during the three weeks before chicks were moved to the rearing compartment at 21 ± 1 (20–25) days of age. In the rearing compartment, indoor temperature was 18.2 ± 1.6 (16–20) °C on all farms except one, where smaller mobile houses were in use, in which indoor temperature was not regulated and was thus similar to the outdoor temperature. Chickens were allowed 8 (five farms), 8.5 (one farm), 9 (one farm) or 12 (one farm) consecutive hours of nocturnal rest without artificial lights. One farmer did not always consider artificial lights to be necessary in summertime, when natural light is ample, so some flocks were reared completely without artificial lighting. On all farms, at least during wintertime, chickens were provided with roughage (silage, hay, lucerne hay and/or straw) ad libitum, distributed loose on the floor or in nets, and/or in small square or large round bales.

Flocks on four of the farms were in general not vaccinated. On the remaining farms, flocks were either regularly vaccinated against Marek’s disease, infectious bursal disease and/or coccidiosis (two farms) or vaccinated sometimes (depending on, e.g., availability of vaccine or disease outbreaks on neighbouring farms) against infectious bursal disease and/or coccidiosis (two farms).

Chickens were allowed outdoor access, most commonly veranda only at first, within approximately one week after moving to the rearing compartment, i.e., at 27 ± 2 (23–30) days of age. In general, free-range access was from 07.30–08.30 h until dark. In summertime, however, chickens on certain farms had continuous access to the veranda (three farms) and free-range area (one farm) also during the night. On all farms, outdoor access throughout the year was largely weather-dependent. The chickens were commonly allowed to free-range from spring (March–May) until autumn (September–November). The pop-holes were commonly closed during outdoor temperatures below 0 °C. Birds on all farms had outdoor access at the time of visit (October).

All organic chickens were slaughtered at the same KRAV-certified abattoir. The entire flock was slaughtered at once on five farms, at 63 ± 1 (62–65) days of age. On the remaining three farms, each flock was slaughtered on two separate occasions (thinning), at 59 ± 6 (53–63) and 66 ± 3 (64–69) days of age, respectively. The birds were manually caught for slaughter transport, one chicken at a time, and placed in crates that held six or seven birds each. Collection and transport were performed at night on all farms. Birds were either collected by trained teams (five farms) or by the farmers themselves (two farms), or a combination of both (one farm).

### 3.2. Housing

All farms but one (which had one arrival and one rearing compartment) had two arrival and two rearing compartments, respectively. Thus, four flocks could be kept concurrently in the latter. On five farms, the houses were new-built upon establishment of organic chicken production, and held two separate arrival compartments adjoining one rearing compartment each (Figure 1). The latter were approximately 647 ± 26 (600–700) m^2^ each and intended for a maximum number of 4600–4800 birds. At placing of the flock, a maximum of 3% surplus is, however, accepted according to KRAV standards, motivated by the fact that it is difficult for the hatcheries to deliver an exact number (normally within −2 to +3%) of eggs or chicks. The order should, however, never exceed 4800 (or the maximum number of birds allowed in a compartment) individuals [27]. Old facilities (previously used for rearing pigs, turkeys or conventional broiler chickens) had been converted on three farms, and further complemented by new rearing compartments in two of these. Rearing compartments on one farm were currently under reconstruction, which is why eight previously used mobile houses were temporarily in place (Figure 2). These mobile houses were approximately 160 m^2^ each and held around 1100 chickens after the birds had been moved from the arrival compartments.

Pop-holes were evenly distributed along the outer long side walls, and connected the rearing compartment to an adjoining veranda (Figure 1), which was present in all but the mobile houses. The number of pop-holes was 8 (six farms) or 10 (one farm) in the rearing compartments on all farms except the mobile houses, which had two pop-holes on each long side wall. Along the entire length of the veranda there was a single curtain (Figure 1), which was automatically or manually regulated for free-range access. Only the mobile houses had no windows for provision of natural light, but instead curtains (for manual regulation of ventilation) at ground level on each side of the house, which also functioned as light inlets.

The arrival and rearing compartments were each completely separate units, with no direct contact between the different flocks. Hygiene barriers were located outside the entrance of each compartment, as well as at the main entrance (Figure 1). Each compartment was emptied, cleaned and disinfected between flocks.

#### Indoor Environment

Bird density during farm visits (with reference to the number of birds and mean body weight at the time) in rearing compartments (veranda excluded) was 6.7 ± 0.5 (6.0–7.4) birds per m^2^ or 17.1 ± 1.9 (14.3–19.3) kg per m^2^. There were commercial nipple drinkers with drip cups (Big Dutchman^®^ Top Nipple/Top Nipple Orange [28] on five farms, missing information for three farms) and commercial feeding pans (Big Dutchman^®^ FLUXX 330/360 [28] on five farms, missing information for three farms) on all farms. The total number of drinkers in each of the observed rearing compartments (mobile houses excluded) was 555 ± 145 (411–759) and the total number of feeding pans was 131 ± 25 (102–179). There were 9.1 ± 2.1 (6.3–11.7) birds per drinker and 37.7 ± 6.6 (26.8–47.1) birds per feeding pan, with reference to the maximum number of birds allowed in the rearing compartments.

Wood shavings were used as litter material on three farms. Wood shavings and peat (two farms), wood shavings and straw (one farm), and straw only (one farm) were also used. On three farms there was underfloor heating in the rearing compartments, however this was utilised only on two of these farms. Litter quality scores (Figure 3) were not correlated to location in the rearing compartment (*p* = 0.72) or to use of underfloor heating (*p* = 0.16). Because the litter quality assessment criteria (Welfare Quality^®^) were not applicable to the straw used as litter material on one farm, this was excluded from the analysis. The straw used as litter material on this farm was, however, without remarks.

No birds were observed panting or huddling during any of the farm visits. The dust sheet test results showed “minimal evidence of dust” on seven farms and “no evidence of dust” on the remaining farm (with mobile houses and straw as only litter material). During farm observations, there was olfactory evidence of ammonia on three farms, while there was no sensory indication of ammonia on the remaining five farms.

### 3.3. Health

#### 3.3.1. Foot and Leg Health

The total proportions of birds (*n* = 400) with different foot pad dermatitis (FPD) and hock burn (HB) scores are shown in Figure 4. No birds received FPD scores ≥ 3 or HB scores ≥ 2. The prevalence (min-max) of FPD (0–58%) and HB (0–26%) varied widely between the different flocks (Table 2).

The prevalence of HB increased significantly with increasing body weight (BW) (*p* < 0.001), but the prevalence of FPD did not (*p* = 0.64). Age was not correlated with the prevalence of either HB (*p* = 0.17) or FPD (*p* = 0.82). There was no correlation between HB and FPD (*p* = 0.11). Worse litter scores were significantly correlated with a higher prevalence of FPD (*p* < 0.001) and HB (*p* < 0.01). Residuals of the fitted models were found to be adequate. No toe damages were observed in any of the birds.

Chickens were gait-scored indoors (*n* = 200) and, when possible, outdoors (*n* = 151). On two farms, no birds were observed free-ranging. No birds scored ≥4, and all birds with GS 3 were observed on four of the farms during indoor assessments. GS were significantly lower (better) in birds observed outdoors than in birds observed indoors (*p* < 0.001). The proportion of birds observed without remarks was more than twice as large among birds outdoors than indoors (Figure 5a,b).

Indoor GS were significantly higher (*p* < 0.05) in flocks with mean BW ≥ 2594 g (six farms) compared with flocks with mean BW < 2150 g (two farms). No correlations between age and gait scores were found (*p* = 0.11).

#### 3.3.2. Integument

The total proportions of birds (*n* = 400) with different plumage condition and plumage cleanliness scores are shown in Figure 4. No birds received score 3 during this assessment. The prevalence (min-max) of feather damage (0–68%) and dirty plumage (14–96%) varied widely between the different flocks (Table 2). Plumage condition scores from one farm were excluded due to moulting in the flock. Plumage cleanliness scores from another farm were excluded as chickens were wet and muddy due to current weather conditions. There was a significant correlation between dirty plumage and FPD (*p* < 0.01), and between dirty plumage and HB (*p* < 0.001). The prevalence of dirty plumages increased significantly with increasing BW (*p* < 0.001) and with increasing age (*p* < 0.05). Residuals of the fitted models were found to be adequate.

Comb colour was normal in all chickens examined and no comb dehydration was observed. Comb wounds (score 1) were found in 0.5% of all birds. No comb wounds were observed in the remaining birds. Skin lesions were found in 2% (score 1) and 0.3% (score 2) of all birds.

#### 3.3.3. Gut Health

The total proportions of birds (*n* = 400) with signs of enteritis (diarrhoea) are shown in Figure 4. The prevalence (min-max) of enteritis (diarrhoea) (0–44%) varied widely between the different flocks (Table 2). Enlarged crops were found in 1% of all birds.

#### 3.3.4. Body Weight

In each flock, a clinical examination of 50 birds was performed (*n* = 400). Average body weights varied between 1947 and 2800 g (Figure 6) due to age differences between flocks. Flock body weight uniformity, expressed as a coefficient of variation (CV), was 15.0 ± 2.8% (11.8–20.7). The CV was significantly (*t*(5.05) = 3.51, *p <* 0.05) higher (i.e., flock weight uniformity lower) in RR flocks (*M* = 16.5, *SD* = 2.4) than in H flocks (*M* = 12.5, *SD* = 0.7).

### 3.4. Production

The average flock mortality over time estimated by the farmers was 3.4 ± 0.9% (2–5) (Table 3). Mortality reasons reported were presumed heart failure (four farms), predators (three farms, of which one regarded it as a minor problem), general weakness or other unknown reasons in young (age <5 days) chicks (three farms), leg problems in older chickens (one farm), and stargazers, i.e., birds whose head and neck is retracted in an abnormal twisted position due to various aetiologies (one farm). Culling of chickens was mainly due to leg problems, general weakness and reduced growth. Farmers reported that the number of chickens culled varied throughout the production cycle, ranging from none during some weeks to several chickens during other weeks.

Average live BW at slaughter was estimated to be around 2500–3000 g by all farmers except one (who would not make an estimate due to the aforementioned large variations in BW within flocks), with an average daily weight gain of 45–50 (52) g. Average feed conversion rate (FCR) was estimated by the farmers to be approximately 2.0–2.4 kg feed per kg live weight (seven farms) and 2.6–2.7 kg (one farm).

Abattoir records on the flocks observed during farm visits could only be obtained for four of the farms, which is why this information was not included in any further analysis.

## 4. Discussion

### 4.1. Housing and Indoor Environment

Slower-growing hybrids were reared on all farms in compliance with EU regulations [5]. Rearing compartments held no more than 4800 chickens, also according to EU regulations, and bird density at the time of farm visits did not exceed the maximum allowed limits (10 chickens/m^2^ or 21 kg/m^2^) [5]. The latter was, however, calculated based on farmer estimates, and thus holds some uncertainties. The chickens on all farms were provided with natural light, and artificial lights were turned off for at least eight consecutive hours per day [5]. Birds had outdoor access from spring to autumn, and were provided with roughage at least during wintertime when free-ranging opportunities were restricted, as reported by all farmers.

The dust level was low on all farms, and the temperature most likely within the birds’ thermal comfort zone, as reflected by the absence of any birds panting or huddling. Five farmers, however, described difficulties maintaining the indoor temperature, as well as high heating expenses, with decreasing outdoor temperatures. Ventilation may be reduced to conserve heat, but commonly results in a subsequent increase in humidity and, e.g., ammonia concentrations [29,30,31]. The former has been associated with deteriorating litter quality [32], in turn associated with FPD and HB [33,34], of which a higher prevalence has been demonstrated during colder seasons [29,30,31]. The study was performed during the autumn season, thus results regarding litter quality, FPD and HB may have been different in winter or summertime. Only on one farm was litter quality without any remarks. On the remaining farms, at some locations in the rearing compartment, the litter was not dry and loose but compacted and with a solid upper layer, however it was not wet or sticky. Litter of such quality would thus also function poorly as dustbathing material. Maintaining an optimal indoor environment during colder seasons is a well-known issue also in conventional broiler production, further complicated here in housing systems in which the outdoor environment influences the indoor environment through direct communication. Not only can this affect animal welfare in terms of negative effects on physical health, but furthermore, flocks reared in wintertime may never be provided with outdoor access.

### 4.2. Farm Management

The interviews with farmers indicated that there are differences between farmers’ attitudes and motivational factors, but also revealed discrepancies regarding management routines (e.g., vaccination routines, lighting and temperature programmes, veranda and free-range access). This is in contrast with conventional broiler production, which is often very homogenous and where highly standardised management manuals (e.g., [35,36]) are available for the common fast-growing broiler strains. The rearing of slower-growing organic broilers is, however, without any such precise manuals, and appears to be more heterogeneous, even amongst the few farms included in this study. Ideal farm management might not necessarily be achieved through one single uniform approach, but could perhaps comprise different approaches depending on adaptations to the individual flock, farm and farmer. Thus, this heterogeneity amongst farmers and management might perhaps constitute a strength, and should be further investigated from both an animal welfare as well as a production perspective.

### 4.3. Health

#### 4.3.1. Foot Pad Dermatitis and Hock Burns

Foot pad dermatitis and hock burns are painful conditions [37] and are acknowledged welfare issues among commercial broiler chickens [37,38]. Reports on FPD and HB prevalence in commercial slower-growing organic broilers are scarce, however, and a comparison of findings is further hampered by the different scoring systems used [39]. In this study, approximately one-fifth of all birds were observed with minor to moderate signs of FPD, which is notably lower than in other studies on slower-growing chickens with outdoor access reared under research [15] or commercial [39,40,41] conditions. No birds in the present study had severe lesions, again in contrast to the previous studies. Mild signs of HB were observed in 13% of all chickens, whereas the remaining animals assessed had no signs of HB. Similar results were found in earlier studies on slower-growing hybrids, however these were reared under experimental conditions with [15,42] or without [43,44] winter-garden or outdoor access. The absence of moderate and severe HB lesions is in contrast to some earlier findings [42,43], but in agreement with others [44].

Foot pad dermatitis and hock burns have been associated with a number of different factors, which might explain the discrepancies between the present and previous results. Numerous studies demonstrate a significantly increased risk of FPD [31,33,34,45,46] and HB [33,34] when birds are exposed to deteriorated (moist) litter. A significant correlation between poorer litter quality and FPD and HB prevalence was also found in the present study. Outdoor access [41,47] and more frequent range visits [48] have been found to correlate to an increased prevalence of FPD, whereas the opposite has been concluded in other studies [19,40]. More locomotor activity and less contact with litter has been suggested as an explanation for the latter [15,19]. However, outdoor weather conditions are not always optimal, and birds may also be exposed to moist ground as a consequence of high humidity and precipitation [47]. HB prevalence increased significantly with increasing BW, as observed previously [15,31,42,48,49,50], presumably as a result of heavier birds spending more time sitting down with their hocks in contact with the bedding [49]. However, there was no significant correlation between FPD prevalence and BW, which is in agreement with some earlier studies [49,51], but in contrast with others [47,50]. FPD and HB have been associated with different genotypes [31,43,47,49,51,52], with a significantly lower prevalence in slower- compared with fast-growing chickens [43,44,47,49,50,53]. The latter, in combination with more physical activity, lower stocking densities and overall good indoor environments, may contribute to the relatively good foot health in these birds, despite the challenges associated with the Nordic climate.

#### 4.3.2. Gait

Free-ranging and outdoor access have previously been correlated with better gait in fast- [48,54,55] and slower-growing [15,42,54] broilers. In the present study, the GSs in birds observed outdoors were, as expected, significantly lower (better) than in birds observed indoors. Whether birds with lower (better) GSs are more prone to use free-range areas, or whether gait improves as a consequence of outdoor physical activity, is not completely clear. The latter has, however, been demonstrated in previous studies [48,55]. Outdoor access as such, provided that the chickens free-range when given the opportunity, can thus contribute to improved leg health, and consequently better welfare in broilers.

Lameness in broilers has been associated with rapid growth rates [43,56] and high BW [15,42,43,50,56]. Numerous studies show significantly lower (better) GSs in slower-growing chickens compared with fast-growing [44,54,56,57,58]. In previous studies of slower-growing chickens reared under commercial organic [58] or research conditions without outdoor access [42,43,44], no or only a few individuals with the highest (worst) GS were observed. Similarly, in the present study no birds with severe walking impairments were observed. Whether this finding reflects an absence of such walking impairments in the birds studied, or that severely lame animals are appropriately culled by farmers, is, however, unknown.

Gait scoring was performed without confining the birds, to avoid assessing only birds that were less agile or unable to move away, and thus the analysis was limited by the lack of known of individual body weights. However, the GSs for chickens in flocks with a lower average body weight were significantly lower (better) than GSs for chickens in flocks with a higher average body weight. These results must, however, be regarded taking into consideration the low BW uniformity in the flocks observed. Studies show that at least moderate and severe lameness are associated with pain [59,60]. Impaired mobility is, however, also a welfare issue considering that it might hinder the performance of natural behaviour or hamper access to resources. It is an important welfare parameter, with need for improvements to ensure the welfare of the slower-growing broilers within the studied organic production system.

#### 4.3.3. Plumage

Maintaining an intact and healthy integument is essential for bird welfare [61]. The plumage in approximately half of all chickens in the present study was considered slightly or moderately dirty, whereas in the other half it was without remarks. In previous studies on slower-growing hybrids, none of them performed under commercial rearing conditions however, and only one on birds with free-range access [15], the proportions of birds with clean plumage reported have been lower [15,43] as well as higher [42]. No birds were assessed as very dirty in the present study, in accordance with the aforementioned studies [42,43].

The plumage can become wet or soiled with, e.g., litter of poor quality, faecal matter and dirt. With outdoor access, chickens are exposed to variable weather conditions and thus an intact plumage, which is essential for thermoregulation, is particularly important. It has been suggested that outdoor access might improve plumage condition [55,62], but also that individuals with a clean and intact plumage may be more prone to use free-range areas [55]. Plumage cleanliness decreased significantly with increasing BW and age, in agreement with earlier observations [15,43,44]. There was also a significant correlation between dirty plumage and FPD as well as HB. Heavier individuals are likely more prone to become fouled from contact with the bedding and from conspecifics, since it has been shown that these birds spend a large proportion of time sitting down [7]. The relatively cleaner plumage observed in slower-growing chickens compared with fast-growing ones has been explained in the same way [44,57], as the former tend to walk and stand more [6]. Furthermore, access to dust bath materials and the lower stocking density in organic broiler production enables the birds to perform, e.g., grooming and dust bathing behaviour to a greater extent [63], which is important for the maintenance of plumage cleanliness.

The majority of the chickens had no or minor feather damages, as also found in previous studies on slower-growing hybrids [6,53], whereas birds in three particular flocks were observed to have minor to moderate damages. Feather pecking amongst chickens and injuries from objects in the environment can cause feather damage and abrasions or other skin lesions. No particular risk factors were identified in the latter flocks, however. Feather pecking is a major welfare problem among laying hens, in which it has been thoroughly studied, while corresponding research on broilers is limited and does not provide sufficient knowledge of the occurrence of feather pecking and cannibalism. Some of the farmers in this study mentioned injurious pecking as a potential consequence of various flock disturbances, but appeared to be aware of when and how to prevent any outbreaks. Studies comparing slow- and fast-growing broilers report significantly better feather conditions in the former [6,57]. It has also been demonstrated that free-ranging may improve plumage condition and reduce feather pecking in laying hens [64,65,66]. Good management, free-range access and slower-growing hybrids seem thus to be a promising combination for intact plumage, which was predominantly observed in the present study. However, the large proportion of individuals with dirty plumage indicates a need for improvement in factors related to chicken growth and management, in order to ensure good animal welfare in this regard.

### 4.4. Production

#### 4.4.1. Average Daily Weight Gain

Although the growth rate in slower-growing broilers is reduced, this is only in relation to other, more fast-growing hybrids. This is clearly illustrated, e.g., by the current definition of slower-growing used in Germany (cit. [42]), according to which growth rate may not exceed 80% of the daily growth of genotypes bred for top efficiency. Under Swedish regulations, slower-growing hybrids may have an average daily weight gain of 45 g at most [67]. However, the farmers in the present study reported an average daily weight gain of 45–50 g, and sometimes as high as 52 g. The results of this study show that welfare issues attributable to chicken growth rate are still present in slower-growing hybrids, e.g., hock burns and impaired gait, and thus limiting the average daily weight gain is important from both an animal welfare and legal perspective.

#### 4.4.2. Flock Body Weight Uniformity

Flock uniformity is a measure of the spread of live weight in relation to the flock average, often defined as the proportion of birds within 10% of the mean flock body weight. It is commonly expressed as coefficient of variation (CV) [68], which should not exceed 10% according to general recommendations [24,69]. The CV in all flocks observed in this study was notably higher than 10%.

Various management and environmental factors have been shown to influence broiler flock weight uniformity [24], such as management and age of broiler breeders, incubation and brooding conditions, nutrition [70,71,72,73] and feeding management practices [74,75], ventilation, breed [70,71] and health problems. The limited number of studies on flock uniformity in broilers are on fast-growing hybrids [68,69,70,71]. In general, uniformity decreases with increasing age in mixed-sex flocks, due to the faster growth rate in males compared with females [70,76]. Thus, flock variations might become more pronounced in slower-growing broilers, since they are reared for a longer time. It is possible that a flock with high CV in reality consists of two uniform sub-populations, females and males [24]. However, histograms of flock body weights refuted this as an explanation for the large variations in all flocks observed. It has also been suggested that additional floor space at lower stocking densities may allow some individuals to grow more, and hence negatively affect uniformity [69].

Flock weight uniformity was significantly higher in Hubbard compared with Rowan Ranger flocks. Genetic factors, management and age of broiler breeders, egg incubation conditions (RR chicks were hatched on-farm, while H chicks arrived as day-olds) and flock age are possible factors that may have contributed to this difference. High flock weight uniformity is desirable from a production perspective, as poor uniformity has been associated with increased FCR, reduced growth rate and higher mortality, and as large variations in BW may indicate or create health and other welfare issues [68]. Furthermore, anecdotal information suggests that the parent stock hens may not have been sex-separated completely before sexual maturity, and hence dwarf cockerels may have fertilised some of the eggs thereafter collected as broilers. Whether these comparatively very small, but seemingly healthy, birds experience reduced welfare in a flock amongst larger conspecifics is unknown, but they will, however, have negative implications for the workload and economics of production.

#### 4.4.3. Mortality

In general, information about mortality rates and production parameters such as FCR was difficult to obtain from the farmers. Few of the farmers in this study could provide detailed farm and flock information, and most gave only (rather vague) estimates. There was a noticeable discrepancy between calculated flock mortality based on production data and the estimates provided by the farmers, with the farmers generally appearing to underestimate chicken mortality. However, farmer estimates (where relevant) presumably did not include the proportion of eggs discarded or not hatched, while the corresponding calculations did, and thus these mortality rates would appear higher. Moreover, there were some uncertainties in the calculations, since, e.g., the number of birds in the observed flock at the time of visit was often an estimate. One of the main mortality reasons mentioned was predation, which further hampers accurate estimations of the flock size at a particular point in time. The observed flock mortalities calculated post-slaughter were notably higher than the farmer estimates of average, as well as observed, flock mortality. The former was around 7% in two flocks on farms to which day-old chicks were delivered (around 9% in two flocks on farms to which eggs were delivered, mortality rate thus including the proportion of eggs not hatched). This is notably higher than the mean mortality rate reported in other studies (around 1–3%) performed on commercial farms, in both organic slower-growing hybrids [58,77] and conventional fast-growing chickens [58,68,78]. This is also lower than some of the farmer estimates of average mortality rates over time. Further studies should thus be undertaken on Swedish organic broiler farms to scrutinise the mortality rates and reasons, as this may reflect severe welfare issues. Moreover, due to the inclusion of eggs discarded or not hatched in the calculations, flock mortality rates at visit and post-slaughter, respectively, were in general higher on farms with on-farm hatching compared to farms which received day-old chicks. While these figures thus do not allow for a direct comparison of actual mortality between the two systems, they could indicate that receiving the same number of eggs and day-old chicks, respectively, might constitute an economic disadvantage if a proportion (up to 10%) of the former never hatch. Further studies on this, as well as the animal welfare implications of on-farm hatching, are thus required.

### 4.5. Limitations of the Study

To the best of our knowledge, there were only four other organic broiler farms (not included in the study) in Sweden at the time of the study, of which one declined when asked to participate and three were unsuccessfully contacted. Thus, the eight farms visited represent the majority (i.e., 66%) of all commercial organic broiler farms in Sweden, and the results can therefore be considered to provide a solid basis for describing the current situation. The low number of farms is still a considerable limitation, however. The limited number of farms prevented statistical analysis of, e.g., differences between the two hybrids, as it was not possible to tease apart hybrid from other individual farm-related factors, such as, e.g., on-farm hatching (all RR flocks) and the placing of day-old chicks (all H flocks). Moreover, due to the lack of repeated observations, specific management routines or environmental determinants were difficult to statistically analyse in relation to health scores. Repeated farm visits, during a production cycle and in more than one flock, would have enabled observations in a broader context, i.e., different weather conditions or alterations in husbandry routines. Farms were visited in October, and because free-range access was a requirement, flocks were observed as close to slaughter as possible but while birds still had outdoor access. Thus, this resulted in some age variations between flocks, which creates further difficulties in the comparison between these, and ideally birds in all flocks would have been observed at similar ages. On farms where more than one flock was of suitable age, the farmers decided on which flock to be observed. A simple method for randomisation in these cases could have been applied to prevent this potential bias.

## 5. Conclusions

To our knowledge, empirical studies of slower-growing broilers on organic commercial farms are rare. This study provides increased knowledge and a first, although limited, overview of certain health and welfare aspects, housing and management on organic broiler farms in Sweden. Severe health issues were rarely observed during clinical examinations, although birds with minor to moderate lesions and remarks, concerning, e.g., foot and leg health and plumage condition, were found. Higher body weights were significantly correlated to an increasing prevalence of hock burns and dirty plumages. Gait in birds assessed outdoors was significantly better than in birds observed indoors, which may indicate that free-ranging has a positive effect on broiler leg health. In order to better comply with the animal welfare incentives of organic regulations, attention should be paid to the average daily weight gain in the two hybrids studied. Future research should aim at investigating important aspects related to bird welfare, such as the low flock body weight uniformity and the high mortality rates observed in this study.

## Figures and Tables

**Figure 1 animals-10-02098-f001:**
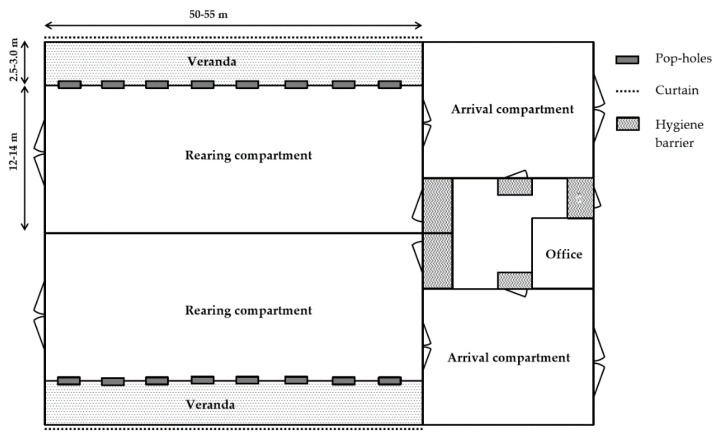
Schematic drawing of typical broiler house with standard rearing compartment dimensions as observed on five Swedish organic farms.

**Figure 2 animals-10-02098-f002:**
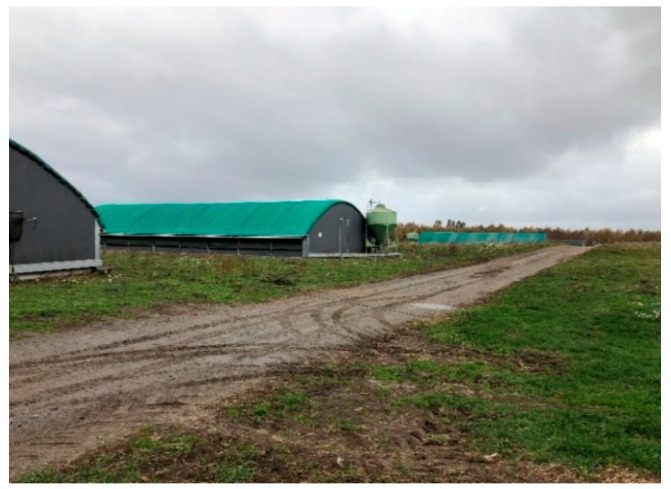
Mobile broiler houses on one Swedish organic farm.

**Figure 3 animals-10-02098-f003:**
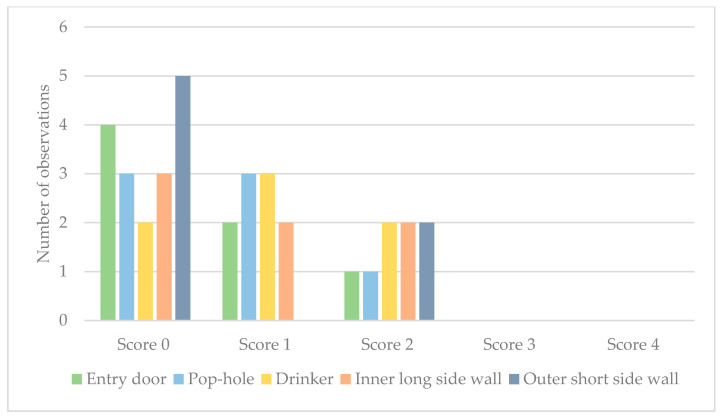
Litter quality assessment scores (Welfare Quality^®^) for Swedish organic broiler farms (*n* = 7): number of observations (*n* = 35) with score 0–4 at five standardised locations in rearing compartments.

**Figure 4 animals-10-02098-f004:**
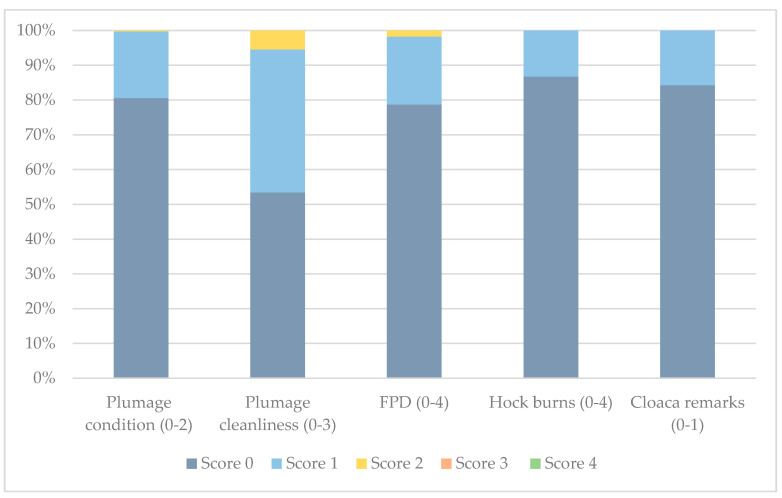
Proportion (%) of chickens (*n* = 400) with different plumage condition, plumage cleanliness, foot pad dermatitis (FPD) and hock burn scores and cloaca remarks (0 = no remarks, 1 = signs of enteritis/diarrhoea) (Welfare Quality^®^) in flocks (*n* = 8) observed on Swedish organic broiler farms.

**Figure 5 animals-10-02098-f005:**
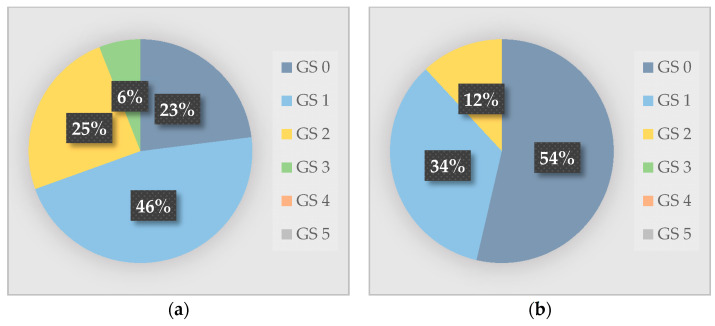
Proportion of chickens (*n* = 300) with gait score (GS) 0–5 (Welfare Quality^®^) on six Swedish organic broiler farms: (**a**) Chickens scored indoors (*n* = 149); (**b**) Chickens scored outdoors (*n* = 151). Two farms were excluded for the comparison, as no birds were observed outdoors in these flocks.

**Figure 6 animals-10-02098-f006:**
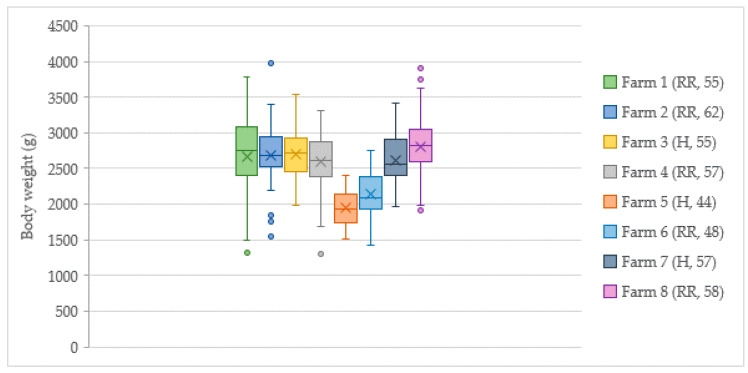
Distribution of chicken body weights in birds assessed (*n* = 50) in flocks on Swedish organic farms (*n* = 8). Hybrid (H: Hubbard; RR: Rowan Ranger) and age (days) within brackets.

**Table 1 animals-10-02098-t001:** Scoring protocol for on-farm clinical examination and gait scoring of commercial organic broiler chickens (including references).

Measure	Scoring
Hock burns ^1,2^	0 (intact skin, no redness)–4 (severe lesions)
Food pad dermatitis ^1,2^	0 (intact skin)–4 (severe lesions)
Toe damage	0 (no damage); 1 (damage ≤ one toe); 2 (damage > one toe)
Plumage cleanliness ^1^	0 (clean)–3 (very dirty)
Plumage condition ^3^ (body and flight feathers combined)	0 (good plumage condition, no or very few feathers damaged);
	1 (completely or almost completely feathered with few feathers damaged, featherless area(s) <5 cm^2^);
	2 (highly damaged feathers, with featherless area(s) ≥5 cm^2^);
	3 (very high degree of damage to feathers, with no or only a few feather-covered areas)
Skin lesions ^4^	0 (no lesions, or <3 pecks or scratches); 1 (≥ one lesion <2 cm or ≥3 pecks or scratches); 2 (≥ one lesion ≥2 cm)
Comb wounds ^4^	0 (no evidence of pecking wounds); 1 (wounds <3); 2 (wounds ≥3)
Comb colour	Normal; red; pale
Comb dehydration	Yes; no
Enlarged crop ^4^	Yes; no
Signs of enteritis/diarrhoea ^4^	Yes; no
Gait (lameness) ^1^	0 (normal, dextrous and agile);
	1 (slight abnormality, but difficult to define);
	2 (definite and identifiable abnormality);
	3 (obvious abnormality, affects ability to move);
	4 (severe abnormality, only takes a few steps);
	5 (incapable of walking)

^1^ Welfare Quality^®^ assessment protocol for poultry, applied to broiler chickens. ^2^ Both feet/legs examined: bird scored according to most severe lesion observed. ^3^ Adapted from Kjaer et al. (2006) [22]. ^4^ Welfare Quality^®^ Assessment protocol for poultry, applied to laying hens.

**Table 2 animals-10-02098-t002:** Proportion (%) of chickens (*n* = 50) in each flock (*n* = 8) with different plumage condition, plumage cleanliness, foot pad dermatitis (FPD) and hock burn (HB) scores and cloaca remarks (signs of enteritis/diarrhoea) (Welfare Quality^®^) on Swedish organic broiler farms. (H: Hubbard; RR: Rowan Ranger).

Farm	Hybrid	Age (Days)	Plumage Condition ^1^			Plumage Cleanliness ^1^			FPD ^2^			HB ^3^		Cloaca Remarks	
			**0**	**1**	**2**	**0**	**1**	**2**	**0**	**1**	**2**	**0**	**1**	**No**	**Yes**
1 ^4^	RR	55	-	-	-	62	34	4	84	16	0	88	12	98	2
2	RR	62	100	0	0	4	76	20	52	48	0	74	26	100	0
3	H	55	84	14	2	78	22	0	82	18	0	96	4	92	8
4	RR	57	32	68	0	52	44	4	98	2	0	100	0	56	44
5	H	44	100	0	0	86	14	0	86	14	0	94	6	80	20
6	RR	48	100	0	0	58	40	2	100	0	0	86	14	88	12
7	H	57	48	52	0	34	58	8	42	44	14	80	20	100	0
8 ^5^	RR	58	100	0	0	-	-	-	86	14	0	76	24	60	40

^1^ Plumage condition and cleanliness: no birds scored 3. ^2^ FPD: no birds scored ≥3. ^3^ HB: no birds scored ≥2. ^4^ Scores excluded due to moulting. ^5^ Scores excluded due to chickens being wet and muddy because of current weather conditions.

**Table 3 animals-10-02098-t003:** Mortality (%) and eggs discarded (%) on Swedish organic broiler farms (*n* = 8). Observed flock mortality (at visit) calculated with reference to original number of eggs/chicks and number of birds in flock at the time of visit (information provided by farmer estimates and/or computer production records). Observed flock mortality (post-slaughter) calculated with reference to original number of eggs/chicks and number of birds delivered to the abattoir.

		Farmer Estimates (%)			Calculated (%)	
	Age (Days)	Eggs not Hatched/Discarded	Average Mortality Over Time (All Flocks) ^1^	Observed Flock Mortality ^1^ (at Visit)	Observed Flock Mortality (at Visit)	Observed Flock Mortality (Post-Slaughter)
On-farm hatching						
Farm 1	55	8–10	4	6	6.7 ^3^	n/a ^5^
Farm 2	62	4	4–4.5	6	6.7 ^3^	8.9 ^3^
Farm 4	57	8–10	4–5	5–6	7.7 ^3^	9.5 ^3^
Farm 6	48	5–6	2–3	2.2 ^2^	10.2 ^3^	n/a ^5^
Farm 8	58	5–10	3.5–4	3.5–4	n/a ^4^	n/a ^5^
Day-old chicks						
Farm 3	55	n/a	2–3	5–6	6.5	n/a ^5^
Farm 5	44	n/a	2	3–4	4.9	7.4
Farm 7	57	n/a	3–4	3.5 ^2^	3.5	7.1

^1^ Unknown whether eggs not hatched/discarded (where relevant) are included in this estimate. ^2^ Information obtained from computer production records. ^3^ Mortality rates including eggs not hatched/discarded due to lack of information on number of chicks hatched. ^4^ Estimate of original and current number of birds too vague for reliable calculations. ^5^ No information on number of birds delivered to abattoir available.

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
