# Peer review of "Bird Health, Housing and Management Routines on Swedish Organic Broiler Chicken Farms"

_animals, 2020, doi:10.3390/ani10112098_

Round 1

Reviewer 1 Report

The authors have made great improvments to the paper.

I want to thank the authors for detailed answers to all of my orignal concerns. I have no further comments. 

Author Response

The authors sincerely thank Reviewer 1 for valuable and very helpful comments. 

Reviewer 2 Report

The revision has significantly improved the publication. Nevertheless, some missing information cannot be added afterwards.

Especially in the areas in the text where changes have been made, there are sometimes too many spaces, but occasionally spaces are missing. Please pay attention to this again.

Many important aspects were supplemented very well in the discussion (e.g. mortality).

The paper provides a first overwiew on organic broiler chicken farms in Sweden.

Special remarks:

L: 275: Headline slipped

L 288, Table 2: Thank you for including Table 2. This gives you a good overview of

how the parameters are related.

Why did you exclude the plumage cleanliness from farm 8. If the birds

are all wet and muddy- that´s a result too

L 295: here you write that 200 animals were examined for gaitscore indoors. In Figure 5a there are only 149 animals? How can this be explained. Please make it clear in the text or the illustration where this difference comes from. Does the difference come from the two farms that were excluded?

L 314: „and“ is missing! …and between dirty plumage and HB…

L 319 and 323: One punctuation mark at the end of the sentence too many

L 330: Does Figure 6 show the body weight oft he birds you examined during your clinical examination or the mean flock body weight?

L 386: Heading 4.2. must go down one line

L 436: Was the light intensity also measured? If it was very dark in the barn, this could also be a reason why the chickens walked less and therefore gave the impression that they walked worse.

L 463: Too much space between words: as well as____higher…

L 472: Too much space between words

Mortality: If 4 slaughterhouses do not report back to the farmers about the delivered animals, how is the billing done there? Does the farmer only get the total weight of his slaughtered animals? In this regard, a paragraph should definitely be added to the conclusions that, with regard to animal welfare aspects, it is very important that selected animal welfare parameters are collected at the slaughterhouses and also reported back to the farmer. Only in this way is it possible to routinely identify conditions relevant to animal welfare on the farms and to take countermeasures.

Author Response

This manuscript is a resubmission of an earlier submission. The following is a list of the peer review reports and author responses from that submission.

Round 1

Reviewer 1 Report

The authors have dealt in depth with the organic production of broilers in Sweden. Since the results will have far-reaching consequences for the management of more and more organic broilers, I would like to suggest to tighten the data and the conclusions based on it.

Abstract:

I would like to recommend to align the abstract and the conclusions closer to the data situation. For example, the comparison is that slower-growing hybrids are better than fast-growing - but this was not the subject of the study. The survey of Swedish organic broiler production is interesting enough to acknowledge that with good management any hybrid can be fattened better or worse. The outcome should therefore not only be reduced to preventing "good enough", but could be a factor-oriented analysis of what leads to good organic production.

Introduction:

The introduction could become stricter at one point or another, e.g. when the importance of the animals' feelings for organic production is discussed - but these feelings are not recorded at all in the study. A slight bias is found throughout the text that conventional broiler production paid "little attention to bird welfare". Although, I know both sides of the story, I would not go so far as to deny that conventional broiler production is interested in welfare. In general welfare is gaining importance, whether for conventional or organic broilers. The expectation that the extensive data available will be used to "identify factors" that lead to good welfare is only partially met in the course of the manuscript.

Methods:

Specific information is missing or is only given in the Range. In my opinion, however, it would be helpful for the entire data situation to specify the mean value with SD/SE or as percentage values with variance specification. It is not quite clear how many animals have been scored at which position in the barn. The biggest question, however, is about the statistics. I would have expected that an ANOVA or similar would be used to get to the bottom of the interrelations between the different factors. It is also not clear to me why a binary data transformation was necessary.

Results:

In general I am missing the information in Average and SD/SE. And basically I am already attached to the genetics of the animals. With the Hubbard animals it concerns once the production level and once the breeder line. However, there is no further mention of them in the manuscript. Here a clarification is urgently needed. The results are very extensive, but also very descriptive. Actually, the data collection should allow for more analytical statistics.

Discussion:

The discussion follows the descriptive presentation and could, in my opinion, be significantly streamlined. Parts of the text about topics that do not belong to the survey or survey could be shorter and individual anecdotes could be left out. The results are good enough to be compared with the existing literature and to reflect the status quo in Sweden. In my opinion, however, the aim should be to identify more clearly the factors that lead to increased animal welfare in organic production.

Specific comments:

L14: appeared to be - yes or no?

L17-18: minor to moderate - for a laymen hard to understand.

L20: fast-growing hybrids were not part of this study

L22: Avoiding welfare issues is not (only) a matter of rapid growth but also of a wise management

L26: 8 out of 12 paricipated the study

L31: Dirty plumage is a broad term, here minor or moderate would describe it in more detail

L48: instead of "without the opportunity" I would suggest "with limited access to"

L56: dustbathing is not necessary an outdoor behaviour

L77: Expecting correlational conclusions based on te identification of factors

L86: Do you think that the participation of certain companies constitutes a bias (evaluation part)?

L88: Throughout the ms, please provide mean oder median and a measure of variance accordingly.

L88: Are there regulations of a minimum days of life in organic broiler production?

L98: average number of flocks and rearing compartments per farm?

L100: Do you think that the farmer's selection of a certain flock constitutes a bias (evaluation part)?

L107: Did every barn had a veranda?

L122: This is in contrary to L120

Table 1: add "including references" to the legend

Table 1: I do not understand footnote 2

L131: were examined birds catched at the same beforementioned locations?

L135: Do you think that you action indoor contributed to the number of animals outside as these were performed last?

L143: I do not understand why raw data were grouped or pooled

L153: were there FPD scores above 2 or not?

L163: I would expect FPD to have a "large impact" on gait score, too.

L163-164: Not sure what the correlation of body weight and outdoor use is. You should state exclusion criteria, why wich farm was in- or excluded from the analysis.

L178: Is there an impact of previous experience on the welfare level?

L184: on-farm hatching vs. day-old chicks is a really hard bias in the data set which should be elaborated in the "study limit" section (including hatching over 5 days and flock uniformity).

L189 and L192: were workload and behavior part of the study (questionaire)?

L196: As the lighting program is crucial, please give information on mean /SD, SE

L207: Did farmers know about conventional management guides?

L207/L209: One farmer or some flocks, didn't you visit just one flock per farmer?

L213: Are there no obligatory vaccinations by law?

L214: Why is Marek vaccinated regularly? Isn't it a one-dose infection?

L222: Were organic broilers produced from November to March without free-range access?

Table 2: line 8, why in brakets?

L260: Any information on the length of the long side available?

L266: Any information on LUX or percentage of daylight related to the barn floor?

L280: With reference to the maximum number allowed or the maximum number housed?

L282: only litter material vs. L283 straw only

L287: What was the score (rather than no remarks)?

Table 4: It is extremly hard to imagine what the number or bales means for each single bird in the barn at a given time.

L310-311: Width/heidth?

L321: What is given, mean oder median and SD or SE? Since the weight is highly correlated with the bird's age, does it make sence to provide a common mean although birds age differed 11 days? Same is relevant to figure 5. Would an index of age and weight make sense?

Table 4: Marked cells unclear

L352: Provide percentages to compare flocks

L392: Since the FCR is very important for any decision for or against a genotype or production system, one sentence containing the word approximately might not be sufficient.

L402 - 409: The intro into the discussion is hard to follow. Starting with prioritising buisiness aspects to environmental enrichment and mortality. Please restructure the discussion part to make it more rigor and aligned to the results. Some parts are just narrative and far away from the data set.

L422 - 429: First it is mentioned that homogenous flocks and management guides are desirable to slower growing genotypes, later it is stated that different approaches might be necessary for individual flocks. What should organic management be, homogenous and standarised or heterogenous?

L441: I do not know which enrichments has been used to what extent by the birds. And if so, which were attractive?

L442: "not enough". How is setting a reference line?

L456: Provide references

L461: Is winter-farden and veranda the same?

L501: dustbathing was not part of the survey

L555: What about sex differences in scoring?

L560: was this true for one of the farms?

L570: Did you analyse FCR?

L587: redundant to L588, which negative environmental consequences are meant?

Appendix A: some parts were not analysed, e.g. age of installation of environmental enrichment, expected average slaughter weight or feed specifics????

Reviewer 2 Report

Review of «Bird health, housing and management routines on Swedish organic broiler chicken farms”

General comments:

The topic of your study is interesting and important, since the clear majority of broiler research focus on commercial non-organic production.

However, I have some major concerns regarding the set-up of your study. Your study does not have a control group. A group of fast-growing hybrids reared in the same conditions would have allowed a lot more confidence in the claim that: “The results confirmed that slower-growing hybrids are more suitable for organic farming than fast-growing hybrids”. I cannot see that neither the study design nor the results back this claim. Greater caution is need for several of your claims.

In addition, the material and methods need elaboration. I am not convinced that 25 birds for gait scoring is sufficient. Clinical observation of birds only near the wall may give you a biased selection of animals. Are you certain that this is representative?Also, you examined birds between 44 and 62 days of age. This is a rather broad span for birds that are slaughtered rather young. Could this have affected your results?

The Material and methods along with the result section are in need of major editing, re-structuring and should be made briefer.

You have a lot of data; the manuscript would be much more valuable if you looked for associations between your findings. For instance; did variable the stocking density on the farms affect any of the results (for instance gait scoring or cleanliness)? Any association between enrichments and gait scoring or clinical observation?

To the discussion: not all of your results are discussed. You need to structure the result-section better and discuss each of the major findings.

Overall, I get a bit confused since there are so many factors included in the same study and not all of them are linked. It would be interesting if you could look at the results in a broader perspective. Could it be an idea to separate the results into several papers, one on bird observations and one with farmer attitudes ++? Because at the moment they don’t flow well together.

Abstract:

Line 26-27: Please state bird age during visit and hybrid.

Line 32: please include GS category for mild to moderate impairment

Line 34: what you have presented of results in the abstract does not back your statement that “slower-growing hybrids are more suitable for organic farming than fast-growing hybrids”. Include more results to back your statement or rephrase.

In addition; you have not included a fast-growing hybrid, under the same conditions, as a control group. Based on this I cannot see that you can make such firm conclusions. I suggest that you rephrase.

Line 35: please state what the remaining important welfare challenges are. Lameness? Dirtiness? Insufficient perch space? Likewise, for the next line (36): state the minor and moderate health problems that must be rectified.

Introduction:

Line 44- 46: please add reference to this statement. In addition, I suggest you replace “little” with “less” in line 46.

Line 46-47: please give examples of these health problems.

Line 51: please state the other three dimensions of sustainable animal farming.

Line 66-67: “Moreover, research on slower-growing hybrids has demonstrated that organic broiler farming may involve animal welfare issues and other challenges”. Please give examples.

Line 68-69: “Numerous studies show that, despite their physical ability, only a small proportion of the flock ranges freely when given outdoor access and most chickens remain in close proximity to the house.” Please refer to some of these numerous studies. And is it a welfare challenge that the birds remain in close proximity to the house?

Line 73-74: please add reference to the statement that “knowledge of bird welfare, production and management on farms is still limited.”

Materials and Methods:

A section that addresses the birds would be helpful. Please include: hybrid, slaughter age, daily weight gain, slaughter weight, mortality rate, feed and water supplies (ad libitum? Manufacturer?), flock size, stocking density. The majority of section 3.1 and 3.2 should be moved to M&M-section. 

Line 87: please specify where in Sweden the farms are situated.

Line 89: include slaughter age for the flocks

Line 110-111: the behavioural observations are presented elsewhere, as in another paper? Including the results from panting and huddling birds? Then I suggest you remove all reference to behavioural observations in the current paper. You also mention these observations in your abstract. If you don’t present the results in this manuscript it will not give the reader any information at all to know that you have done the observations.

Line 113: you gait scored 25 birds at each farm? I find this to be a low number. Why did you choose this set up versus 50 birds for clinically examination?

Line 114: please include more information about the gait score scale. What are the criteria for each category? A table would be most helpful.

Line 123: “confined against the wall”, does this mean that you did all clinical examination on birds located along the sides of the building? Is this representative for the flock?

Table 1, plumage condition: is the plumage development equal for birds at 44 and 62 days? Could featherless areas have a connection to the age or the breed of the bird?

Results:

Line 174-178: this belongs to the Material and Methods-section.

Section 3.1 and 3.2: the majority of this belongs to M&M-section

Line 191-194: should be moved and discussed under Discussion.

Line 237-246: did you investigate the birds after your farm visit? Or did you include results from the slaughter house? If not, I suggest you delete this part as it adds no value to the manuscript.

How was the assessor trained in gait scoring and clinical examination? Please add this information.

Line 318, section 3.3: this is where your results start. I suggest you make the above sections significantly shorter and move the information to M&M.

Line 363-363: “GS were significantly lower (better) in birds observed outdoors than in birds observed indoors”. This is perhaps one of your most interesting results. Suggest you lift this, both in the discussion, conclusion and in the abstract.

Figure 7: the colour of GS 0 and GS 4 is very alike. I almost interpreted your figure wrongly. Suggest you alter one of the colours.

Discussion:

Overall, I get a bit confused since there are so many factors included in the same study and not all of them are linked. It would be interesting if you could look at the results in a broader perspective. Could it be an idea to separate the results into several papers, one on bird observations and one with farmer attitudes ++? Because now they don’t flow well together.

Not all of your results are discussed. You need to structure the result-section better and discuss each of the major findings.

Line 397: since no carcass rejection results or autopsy are included, I do not agree that you “provide an overview of bird health”.

4.1: did you find any association between farmers response and the observations from the birds, like gait scoring or cleanliness? In addition, the farmers responses should be a part of the result-section.

Line 441: which enrichments were well used? I cannot see this information anywhere. Can you from your results regarding environmental enrichment make any conclusion regarding bird health or welfare?

4.2.2: any a association between plumage, age, environmental enrichment, hybrid, gaitscore, use of the outdoor space or farmers attitudes?

4.2.3: were the GS results affected by the age of the flock at the time of the farm visit? Or by the farmers attitudes? Or hybrid? You should discuss the the age of investigation; much happens between 44 and 62 days for a broiler chicken.

Line 582: “However, the farmers in the present study reported an average daily weight gain of 45-50 g, and sometimes as high as 52 g. These results confirm previous findings, and demonstrate that welfare 583 issues attributable to chicken growth rate are still present in slower-growing hybrids.” I cannot see that the result back the claim in the second sentence. Please rephrase.

Line 594: this should be 4.4, not 4.3. I do not think you have addressed all the limitations of your study.

609-611: you did not include a control-group, so you cannot claim that this confirms a better welfare. Indicated, perhaps?

Reviewer 3 Report

As the title suggests, it is a recording of the current situation regarding health, housing and Management routines on Swedish organic broiler chicken farms.

One can assume that the aim was not to carry out a scientifically founded study, but only to ascertain how the animals are kept in this production type and in what health condition the animals are under the respective housing and management conditions in one fattening trail.

Overall, it would have been desirable to show the animals in the illustrations according to their housing system and, above all, according to genetics. This is the only way to see whether one of the genetics in a certain husbandry system may show better health etc. The data can only be used to a limited extent to find out which genetics or which husbandry method is more suitable for Swedish organic broiler chicken farms.

Were the requirements of the EU organic regulation met on the farms visited? A small paragraph about the legal requirements for ecological farming in the publication would be useful. Required Enrichment, stocking density, outdoor area etc.

As the authors themselves already under point 4.3. have written, the study shows large gaps in order to finally be able to make a valid statement. With 8 farms it should have been possible to visit the farms at least more than once at the end of the fattening period. So you have the great passage effect of a fattening trail, which can greatly influence the results achieved.

It is also not clear in which season the farms were visited. Was the study carried out on all farms in the same season with the same climatic conditions? If this didn´t happen, it can also have a major impact on the results.

Overall, the publication can be viewed as a snapshot at Swedish organic broiler chicken farms, but the study does not provide clear scientific results on animal health in my view. It can only provide information on any weaknesses that may currently exist.

Special remarks:

L 19:  why was the quantity insufficient?

Were the requirements of the EU organic regulation met?

L 33/34: see comment above

L 88: Why did the animals show such a large age difference of 44-62 days? Gait score should be assessed no earlier than the last 7 days before slaughter. Ideally shortly before slaughter, as the animals show also a big weight gain at the end of the fattening period.

L 98-111: The assessment of the litter quality is missing. What scores were there based on the classification?

L 125: Why weren't the same animals examined for which the gaits core was recorded? It would have a much better informative value if you also had the data on weight, FPD and HB for the individual animals.

L 165: Why were all gaits scores of ≥ 1 summarized? That makes no sense at all, because in a 6-stage system there are great differences in the animals' ability to walk. The scores should be considered individually.

L 167: Why was it not possible to obtain an individual body weight for the examined animals? Average weight in the barn is not very meaningful in this context. Was the uniformity of the herd assessed, too? If yes, how?

L 169: Gaitscore ≥ 2 were pooled. In line 165, it says: scores ≥ 1 were pooled. Which is right? Please make it clear in the text which scores were pooled and, above all, why?

L 276/277: Why is it not clear which drinking and feeding system was used on 3 farms?

L 282: The litter is known from only 7 farms. What bedding was used on the 8th farm?

L 289, Fig. 3: The scores for litter quality are not known and should be added to the material and methods section.

Fig. 3: As I understand it, the illustration makes no sense because we have different housing systems and also different litter materials. It would be better to show the stalls separately according to the type of stall and also according to the litter. Here it is not clear in which stall and which litter or at which place in the stall problems with the litter could occur. It´s all mixed up now.

L 297:

Why was the ammonium level not measured? Evaluation solely based on human senses is not scientific, as there are different individual limits of perception and, moreover, it must always be measured at head level of the animals, and at different points in the barn to get a meaningful result.

L 299: What is a high ammonia level?

L 300: Panting can also result from excessively high temperatures in the stable. If animals show shortness of breath due to excessively high ammonia levels and / or sit huddled together (huddling), animal welfare-relevant ammonia values have long been exceeded and action should have been taken much earlier.

L 331: In Figure 6, too, it would have been better if the illustration had been shown separately according to genetics and housing systems.

L 342: Integrate genetics on the respective farms into the table.

Table 4: It seems as if 100 animals were examined per farm and not just 50? Please represent exactly.

L 346: Here it must be added that you did not have the individual animal weight, but only the average herd weight.

L 348: What is worse litter score? There is no definition for it. Please add it in material and methods.

L 380: Table 5: Why is the OF mortality (post slaughter) sometimes twice as high as the OF mortality (current)? Some farms were visited at the end of the fattening period? The table should also indicate how old the animals were when the farm was visited and at how many days the animals were slaughtered.

L 400: In order to be able to answer this question satisfactorily, one would have to look in more detail at the results (housing system, genetics, litter ...). Additionally, the individual animal weight and the measurement of ammonia would also have been important.

L 412: flock mortality: at least at the companies that get the chicks delivered as one-day-old chicks, the loss can easily be calculated because you know how many chicks will be delivered and how many will ultimately arrive at the slaughterhouse. Not knowing the exact mortality of your herd represents an animal welfare problem as such. There may be a few unexplained "losses" in free-range farming, but not to the same extent as it seems to be the case here on the farms!

L 468: as mentioned before please add the scores of litter quality to the material and Methods

L 527: This sentence is confusing. “Better” should already be integrated into this sentence or the sentence will be reformulated.

L 603: but a descriptive representation of the different genetics would be possible
